# A Modular Deep Learning Pipeline for Cell Culture Analysis: Investigating the Proliferation of Cardiomyocytes

**Lars Leyendecker**[*1]                                    LARS.LEYENDECKER@IPT.FRAUNHOFER.DE
**Julius Haas**[*1]                                         JULIUSHAAS91@GMAIL.COM
**Tobias Piotrowski**[1]                                    TOBIAS.PIOTROWSKI@IPT.FRAUNHOFER.DE
**Maik Frye**[1]                                            MAIK.FRYE@IPT.FRAUNHOFER.DE
**Cora Becker**[2]                                          CORA.BECKER@UNI-BONN.DE
**Bernd K. Fleischmann**[2]                                 BERND.FLEISCHMANN@UNI-BONN.DE
**Michael Hesse**[2]                                        MHESSE1@UNI-BONN.DE
**Robert H. Schmitt**[1,3]                                  ROBERT.SCHMITT@IPT.FRAUNHOFER.DE

[1] *Fraunhofer Institute for Production Technology IPT Aachen*

[2] *Institute of Physiology I, Life and Brain Center, Medical Faculty, University of Bonn*

[3] *Laboratory for Machine Tools and Production Engineering WZL of RWTH Aachen*

**Editors:** Under Review for MIDL 2022

## Abstract

Cardiovascular disease is a leading cause of death in the Western world. The exploration of strategies to enhance the regenerative capacity of the mammalian heart is therefore of great interest. One approach is the treatment of isolated transgenic mouse cardiomyocytes (CMs) with potentially cell cycle-inducing substances and assessment if this results in atypical cell cycle activity or authentic cell division. This requires the tedious and cost intensive manual analysis of microscopy images. Recent advances have led to an increasing use of deep learning (DL) algorithms in cellular image analysis. While developments in image or single-cell classification are well advanced, multi-cell classification in crowded image scenarios remains a challenge. This is reinforced by typically smaller dataset sizes in such laboratory-specific analyses. In this paper, we propose a modular DL-based image analysis pipeline for multi-cell classification of mononuclear and binuclear CMs in confocal microscopy imaging data. We trisect the pipeline structure into preprocessing, modelling and postprocessing. We perform semantic segmentation to extract general image features, which are further analyzed in postprocessing. In total, we conduct 173 experiments. We benchmark 18 encoder-decoder model architectures, perform hyperparameter optimization across 28 runs, and conduct 127 experiments to evaluate dataset-related effects. The results show that our approach has great potential for automating specific cell culture analyses even with small datasets.

**Keywords:** Cellular Segmentation, Deep Learning, Cardiovascular Disease

## 1. Introduction

Cardiovascular disease is a leading cause of death in the Western world. Since the regenerative capacity of the adult mammalian heart is limited, we are exploring novel strategies

---

[*] Contributed equally

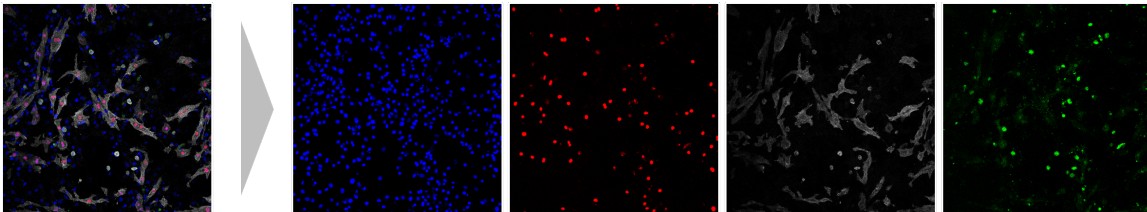

Figure 1: Confocal microscope image and individual 2D-projected fluorescent channels *DAPI (blue, cell nuclei), Cy3 (red, CM nuclei), Cy5 (gray, α-actinin)* and *EGFP (green, cell cycle activity)*

to enhance endogeneous cardiac repair. One approach is to reactivate the cell cycle activity and the cell division of cardiomyocytes (CMs). Cardiac cell biology is complex, as within a few days after birth, CMs switch from hyperplastic to hypertrophic growth. This is accompanied by atypical cell cycle activity, namely acytokinetic mitosis (mitosis without cytokinesis), resulting in binucleated CMs, and endoreplication (mitosis without cytokinesis and karyokinesis), leading to increase in DNA content (polyploid) CMs. In the literature, some reports (Hosoda et al., 2009; Senyo et al., 2013) claim very high rates of CM proliferation and division, and this appears to be due to the technical difficulties to discriminate between atypical cell cycle activity and cell division.

To distinguish between these variations of the cell cycle and authentic cell division, we have established a test system based on double transgenic αMHC-H2BmCherry/CAG-eGFP-anillin mice. αMHC-H2BmCherry labels CM nuclei by the red fluorescent protein mCherry, whereas eGFP-anillin enables to monitor cell cycle activity and progression with high spatial and temporal resolution in M-phase (Hesse et al., 2012). This double transgenic system enables to unequivocally identify CM nuclei and cell cycle variations (Raulf et al., 2015; Hesse et al., 2018). Isolated CMs from hearts of this double transgenic mouse line can be used to screen for substances that are able to bring adult CMs back into the cell cycle and to promote cell division. Such substances would have a strong translational impact as they could increase the regeneration potential of the heart upon injury. Cellular readouts are the number of nuclei for distinguishing mononuclear and binuclear CMs (recognized by mCherry expression), and their cell cycle status, identified by eGFP-anillin+ signals. CMs are isolated by heart dissociation, cultured and treated with several potentially cell cycle-inducing substances for three days followed by fixation and staining with an antibody against α-actinin and Hoechst nuclear dye. Images of CMs are generated by using an inverse confocal fluorescence microscope (Nikon Eclipse Ti2/A1R HD25) with 20x air objective (CFI90 20XC) and four channels for DAPI, EGFP, Cy3 and Cy5 (see Figure 1). The resulting dataset consists of 32 images in $1024 \times 1024$ pixels resolution. Each 3D input image is composed of four fluorescent channels, with six z-layers sampling the image channel in 0.8 $\mu m$ steps, which are transformed into a 2D representation using maximum intensity projection for dimensionality reduction. The input data comprises four test series with eight images per series ($4 \times 8 = 32$). Each series represents a set of images of postnatal CMs, that have been induced with various substances (DMSO, SB63, SB80, WS6), which may effect the cell cycle activity.

Such analysis tasks are monotonous, cost intensive and subjectivity impairs the comparability of the results. In recent years, deep learning (DL) has been increasingly used to automate biomedical image analysis. However, research laboratories often deal with a variety of such specific analysis tasks. Small experiment scales and high costs for manual expert labeling result in small dataset sizes. Varying imaging techniques and microscope models further affect data comparability across laboratories. Oftentimes, the evaluation of multiple metrics is required (e.g., cell number, distribution, activity, size). A data-intensive end-to-end approach is therefore not only ill-suited but also too inflexible for the diversity of cell culture analysis. In this paper, we therefore develop a modular, data-efficient and flexible DL-based pipeline for analysing CMs in complex multi-cell classification scenarios. We trisect its structure into preprocessing, modelling and postprocessing. The pipeline sequentially performs data preprocessing, semantic segmentation, classification and quantification to determine the distribution of mononuclear and binuclear CMs. To identify the best-performing pipeline configuration and to investigate dataset-related effects, we benchmark 18 encoder-decoder model architectures, perform hyperparameter optimization, and conduct 127 data-centric experiments.

## 2. Related Work

High-throughput analysis of microscopic images has long been an investigative topic in cell biology. Applications comprise image classification, image segmentation and object tracking. Early approaches to image segmentation include the morphological analysis and pattern recognition of hand-crafted features. Relevant work includes the edge-based (Marie-Pierre Dubuisson et al., 1994), threshold-based (J.B. Xavier et al., 2001), and the region-based approach for automatic segmentation of cellular structures (E F Battenberg and Ilka Bischofs-Pfeifer, 2006). Due to the advances in recent years, DL algorithms are increasingly replacing traditional image processing techniques. An overview of the state-of-the-art DL-based cell culture analysis is provided by (Moen et al., 2019; Erik Meijering, 2020; Cheng et al., 2021; Deshpande et al., 2021; Wang et al., 2018). For analyzing images with multiple cell instances, one approach is to transform the multi-cell classification problem into a single-cell one. For this, the original image is segmented so that there is only one cell instance on each subimage. Single-cell classifiers are trained on these subimages in a supervised manner. (Shifat-E-Rabbi et al., 2020) compare numerical feature extraction, end-to-end classification with neural networks, and transport-based morphometry on different cell types. (Oei et al., 2019) propose the application of convolutional neural networks (CNN) in single-cell classification based on actin-labeled fluorescence microscopy images that outperform human experts. Another approach to both single and multi-cell analysis is to perform pixel-level classification to semantically segment different cell types or areas. Most studies on this approach use end-to-end encoder-decoder model architectures, such as UNet for biomedical image segmentation (Olaf Ronneberger et al., 2015; Falk et al., 2019), Feature Pyramid Networks (FPN) for object detection (Tsung-Yi Lin et al., 2016) or LinkNet (Abhishek Chaurasia and Eugenio Culurciello, 2017). (van Valen et al., 2016) perform single-cell image segmentation by converting it into an image classification problem and thereby show that CNNs can accurately segment the cytoplasms of bacterial cells and mammalian cell nuclei from fluorescent images. In contrast, instance segmentation tries to

identify each instance of a cell in an image (Moen et al., 2019). Instance segmentation builds upon DL-based object detection techniques such as Faster R-CNN (Ren et al.), RetinaNet (Lin et al.) and Mask R-CNN (He et al.). Furthermore, conditional generative adversarial networks show promising results in nuclei instance segmentation (Mahmood et al.) and cell tissue segmentation (Häring et al., 2018). However, the present cell structures of CMs are characterized by a large number of intermixed mononuclear and binuclear cells with identical nuclei geometry. Because the determination of binuclear cells requires the consideration of multiple nuclei, single-cell classification is not applicable.

## 3. Methodology

We develop a modular DL-based pipeline and explicitly trisect its structure into preprocessing, modelling and postprocessing (see Figure 2). Data preprocessing, semantic segmentation, classification and quantification are sequentially performed to determine the distribution of mononuclear and binuclear CMs. The modular design enables the implementation of application-specific postprocessing procedures with minimal effort and thereby introduces a considerable level of flexibility. Raw microscopy images are preprocessed and enhanced

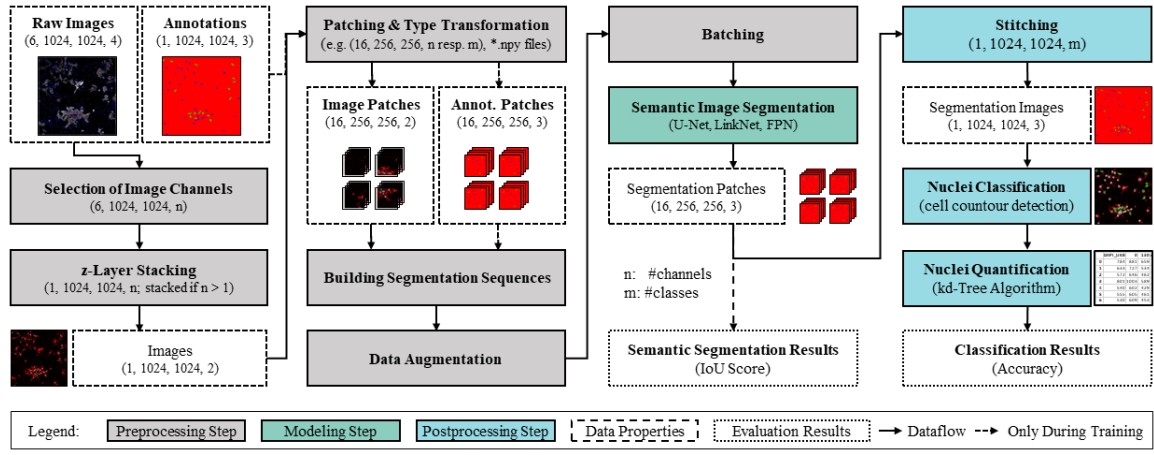

Figure 2: Modular image analysis pipeline constituted of preprocessing, modelling and postprocessing with the corresponding data properties

to achieve optimal modeling results. First, raw images and annotations are converted from the proprietary file format into four-dimensional arrays. Image enhancement includes the selection of $n$ out of four fluorescence channels, processing and stacking the z-layers of each image channel into normalized 2D grayscale images. We design all subsequent operations to be conditional on these dimensions, so that the pipeline can cope with varying image dimensions. On the advice of the experts, only 2D-projected *Cy3* and *Cy5* channels are used, as they contain the relevant spatial features. Images and annotations are split into quadratic patches of subordinate size. Thereby, no informational content is lost due to dimensionality reduction while an efficient network size can be maintained. After sequencing the cached patches for efficient memory usage, oversampling and image augmentation techniques can be applied to artificially increase the amount of data. Augmentation encompasses randomly

sampled spatial and pixelwise transformations, such as *vertical and horizontal flips*, *random rotations*, *random zoom*, *random brightness*, *random blur*, and *Gaussian noise.*

In the modeling step, we use a symmetrical convolutional encoder-decoder model architecture for semantic segmentation and choose the intersection over union (IoU) as evaluation metric and optimization criterion. In postprocessing, the patches are stitched back together to obtain the dimension of the input image. Based on the segmentation image, postprocessing functions aim to extract analysis-specific information. For the quantification of CM cell nuclei, we first apply a combination of morphological opening and closing operations to remove pixel misclassifications. Subsequently, we classify the nucleus types by detecting continuous points along the cell boundary within the inferred segmentation masks. A set of specific postprocessing steps is required to assess the correct instance count of binuclear CMs (i.e., CMs in the final mitotic phase before cytokinesis), as binuclear instances during cell mitosis are represented either as a single eight-shaped contour or two separate nuclei within the same cell body (see Appendix A, Figure 6). This leads to a distortion of the quantification result, as some binuclear contours are counted as one instance, while others are incorrectly counted as two. First, we connect binuclear instances composed of two shapes, utilizing a kd-Tree algorithm (Bentley, 1975) to list all pairs of contours within the range of a defined distance threshold. Next, we apply a repetitive redundancy filtering to ensure that each contour is part of only one binuclear instance. In case of redundancy, we obtain only the connection with the smallest distance in between the two contours. We repetitively execute the filtering procedure until all contours meet the criteria.

## 4. Experiments

To identify the best performing pipeline, to investigate data-related effects and to evaluate the performance on CM classification, we conduct a series of experiments (see Figure 3). We benchmark 18 different randomly initialized encoder-decoder model architectures for semantic segmentation. Having identified a best-performing configuration, we perform a hyperparameter study. In 127 concluding experiments, we investigate data-related effects by synthetically influencing the number of images and their qualitative appearance before training. Hereby we seek to gain an understanding about the minimum required dataset size and the relation between dataset size and model performance. In the model-centric

| 1. Model-centric Benchmarking
Identifying the best model architecture | 2. Hyperparameter Optimization
Optimizing model hyperparameters | 3. Data-centric Experiments
Investigating data-related effects |
|---|---|---|
| **Encoder-Decoder Architectures**
U-Net, FPN, LinkNet

**Backbone Implementations**
VGG16, ResNet18, DenseNet121, InceptionV3,
SEResNet18, SEResNeXt50 | **Patch Sizes**
512x512, 256x256, 128x128, 64x64, 32x32
**Optimizer**     **Loss Functions**     **Batch Sizes**
SGD, Adam     CCE, CF, JD     32, 16, 8, 4, 2, 1
**Training-Validation-Split Ratios**
95/5, 90/10, 80/20, 70/30, …, 10/90, 5/95 | **Data Augmentation Techniques**
*none*, *spatial:* vertical and horizontal flips,
random rotations, random zooms-ins, *pixel:* random
brightness, random blur, Gaussian noise
**Number of Input Images**     **Sample Rates**
1, 2, 3, …, 23, 24     16, 8, 4, 2, 1 |

Figure 3: Overview of the conducted experiments with corresponding parameter sets

benchmarking, we train and evaluate 18 different convolutional encoder-decoder model architectures by considering three different model architectures including UNet (Olaf Ronneberger et al., 2015), Feature Pyramid Network (FPN) (Tsung-Yi Lin et al., 2016) and

LinkNet (Abhishek Chaurasia and Eugenio Culurciello, 2017) as well as six backbone implementations including VGG16 (Simonyan and Zisserman), ResNet18 (Kaiming He et al., 2015), DenseNet121 (Gao Huang et al., 2016), InceptionV3 (Christian Szegedy et al., 2015), SEResNet18 (Jie Hu et al., 2017), SEResNeXt50 (Saining Xie et al., 2016). All model-centric configurations are trained with an input patch-size of $256 \times 256$, Adam optimizer, learning rate of 0.0001, categorical cross-entropy loss (CCE) and a batch-size of 4 for a total of 100 training epochs. Due to the small nuclei sizes of 130 pixels on average in comparison to the overall background, precisely tuning patch-sizes and receptive fields is decisive. In preliminary experiments, we therefore investigated different network architectures with a variety of different receptive fields and patch sizes. We perform hyperparameter optimization for the best-performing model by conducting 28 experiments in accordance to Figure 3. As optimization criterion, we choose the test IoU-score (t-IoU). In the concluding data-centric analysis, we analyse model performances with regard to data-related parameters including image counts, sample-rates and data augmentation techniques.

## 5. Results

Figure 7 in Appendix C shows the results of the model-centric benchmarking. Overall, FPN-VGG16 achieves the highest t-IoU of 0.62 for both mononuclear and binuclear CMs, with a prediction time of 14.4 seconds per image. UNet-DenseNet121 yields a t-IoU of 0.60, with 4.0 seconds per image. The LinkNet-ResNet18 performs best achieving a t-IoU of 0.52, with 2.4 seconds per image. The results of the hyperparameter study that was conducted

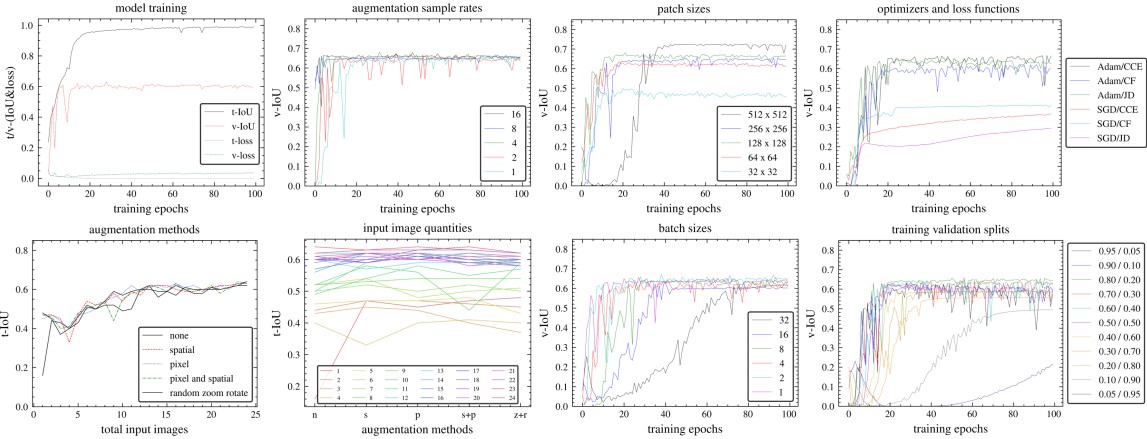

Figure 4: Benchmarking results of the model-centric and data-centric experiments, as well as the hyperparameter study

using FPN-VGG16 as best-performing configuration are shown in Figure 4. Larger patch sizes enhance model performance (see Appendix C, Table 1), but require more training time. A patch size of $512 \times 512$ leads to an t-IoU of 0.65, a patch size of $32 \times 32$ to an t-IoU of 0.51. The study of different batch sizes shows, that a batch size of 2 achieves the highest t-IoU of 0.63 (see Appendix C, Table 3). Generally, different batch sizes do not show a notable effect on the model performance. However, the required training epochs until convergence is positively correlated to the batch size.

The results of the data-centric analysis include the segmentation performance (see Appendix C, Table 5) and final CM classification accuracy (see Appendix C, Table 6) for the specific pipeline (see Figure 2). For each training configuration, the number of full-size input images is iteratively decreased. The row index of each configuration in Table 5 and Table 6 refers to the total number of full-size images (with a training and validation split of 0.70/0.30). Each entry in Table 5 represents the mean t-IoU of the respective configuration across the test set of eight full-size images. Each entry in Table 6 respectively represents the final classification result for all test images, compared to the nuclei count provided by the domain experts from Bonn University. For each augmentation method, the best model performance is highlighted in bold while the best performance across all configurations is underlined. Intuitively, the t-IoU scores in Table 5 are positively correlated with the amount of input images. The highest t-IoU scores across all augmentation methods are in between 23 and 24 input images, with an t-IoU from 0.62 to 0.64. Configurations with six to 22 input images represent 70 % of all model configurations. Notably, several configurations with an input quantity of only twelve full-size images achieve t-IoU scores of up to 0.62, without data augmentation. For 79 % of the input quantity model configurations (19 out of 24 in Table 5), the mean t-IoU for one of the augmentation methods is higher than the t-IoU for the same configuration without data augmentation. For 29 % of the input quantity model configurations (7 out of 24 in Table 5), each of the four augmentation configurations reaches a higher t-IoU than the respective configuration without augmentation. However, the positive effect of augmentation lies within the relative range of 1 - 5 %. For some configurations the performance even decreases with the application of data augmentation.

Figure 5 shows the final classification results of a specimen induced with the *WS6 1:10 A* nutrient, predicted by FPN-VGG16, patch size of $256 \times 256$, Adam optimizer with a learning rate of 0.0001, CCE loss, a batch size of 4 and 100 training epochs. The classification example in Figure 5 includes two missing classifications (red) and two misclassifications (yellow). All other nuclei instances are classified correctly. However, it is important to mention that despite the relatively satisfactory classification result of the example in Figure 5, a notable share of test images across several datasets still showed reasonable divergences compared to the expert labels. Nevertheless, several pipeline configurations in Table 6 achieved averaged test accuracies of up to 0.82 for the classification of nucleus types.

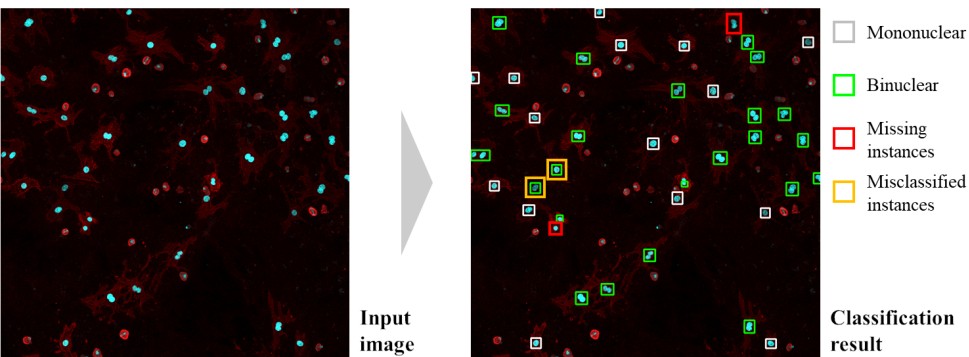

Figure 5: Exemplary end-to-end classification result with FPN-VGG16 model architecture for a specimen treated with *WS6*

## 6. Discussion

The results show that our proposed modular DL-based pipeline for multi-cell image analysis enables biomedical experts to accelerate their research efforts by automating monotonous evaluation tasks. Despite the additional complexity due to the modularity of the pipeline, its modular interchangeability enables easy adoption to new applications. Of all encoder-decoder model architectures, FPN-VGG16 achieves the best t-IoU of 0.62. However, UNet-DenseNet121 achieves an IoU of 0.60, while increasing prediction efficiency by a factor of 3. Although the application of different augmentation methods improves the performance of many models, a distinct correlation cannot be observed. This is reflected in configuration [24, n] in Table 5, where the highest t-IoU of 0.64 is achieved without any augmentation. For all other configurations of that quantity, data augmentation does effect or even deteriorates the performance. Also, the t-IoU scores for different input quantities in Figure 4 show fluctuation but no convergence with the application of augmentation. As expected, input data quantity is positively correlated with segmentation performance. However, configurations with eleven to 14 input images yield only insignificantly lower performances. This indicates the ability of FPN-VGG16 to perform well with limited data. We therefore assume that an increase in training data for this specific application does not necessarily lead to improvements in performance. The t-IoU scores in Figure 4 affirm this assumption, as configurations with input data quantities of twelve images and more do not result in better performance. The final classification results in Table 6 correlate with the IoU scores in Table 5, as the classification is part of the analysis-specific postprocessing. However, this does not apply in cases where a share of the nuclei instances is classified but defectively segmented. This is as a subset of correctly segmented pixels per nuclei is already sufficient for a subsequent classification. A similar IoU score is achieved in scenarios where the majority of instances is correctly segmented and localized, while some instances are missing completely. However, missing instances negatively influence classification accuracy.

## 7. Conclusion

In our work, we propose a modular DL-based image analysis pipeline for multi-cell classification of mononuclear and binuclear CMs. Due to the modularity and tripartition of the pipeline into preprocessing, modelling and postprocessing, we expect it to be easily adaptable to other image modalities and cell analysis tasks. By separating general semantic feature extraction and task-specific postprocessing, we do not pursue a data-intensive holistic end-to-end approach. In an extensive benchmarking of 173 experiments, we investigate the correlations between segmentation model, hyperparameters, dataset properties and analysis accuracy. Thereby, we provide a guideline for the DL-based automation of complex cell culture analyses that can cope with a small number of images and annotations. In future work, we will investigate the transferability of our pipeline and its automatic configurability across various analysis tasks. To further improve the results in the investigation of CM proliferation for cardiovascular research, next steps will consider image transformers, postprocessing optimization and cell activity tracking using the eGFP-anillin fluorescence signal. This would provide a novel and important tool to screen for substances that have the potential to increase the number of CMs and thus can be used in the long term for heart repair upon injury.

## Acknowledgments

The publication was written within the research project AIxCell (IGF 21361 N). The project of the Research Association for Precision Mechanics, Optics and Medical Technology is funded by the Federal Ministry for Economic Affairs and Energy via the AiF within the context of the programme for the promotion of Industrial Cooperative Research (IGF) based on a resolution of the German Bundestag.

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

## Appendix A. Methodology

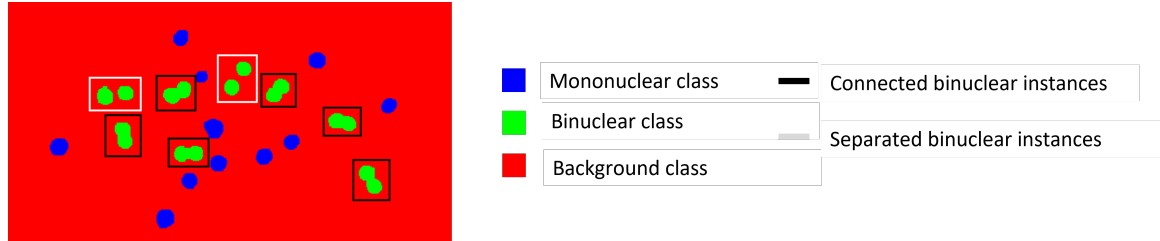

Figure 6: Cardiomyocyte nuclei types in semantic segmentation mask

## Appendix B. Experiments

We train all models on a Xeon E7 remote server, running on Ubuntu 4.15.0-154-generic with 2 x Intel Xeon CPU E5-2680v4 at 2.40GHZ, 264 GB RAM and 2 x 12 GB Titan X GPUs, with CUDA 11.2. All predictions are inferred on a local machine, running on Windows10Pro with Intel Core i5-7300U CPU at 2.60GHz - 2.71 GHz and 8 GB RAM.

## Appendix C. Results

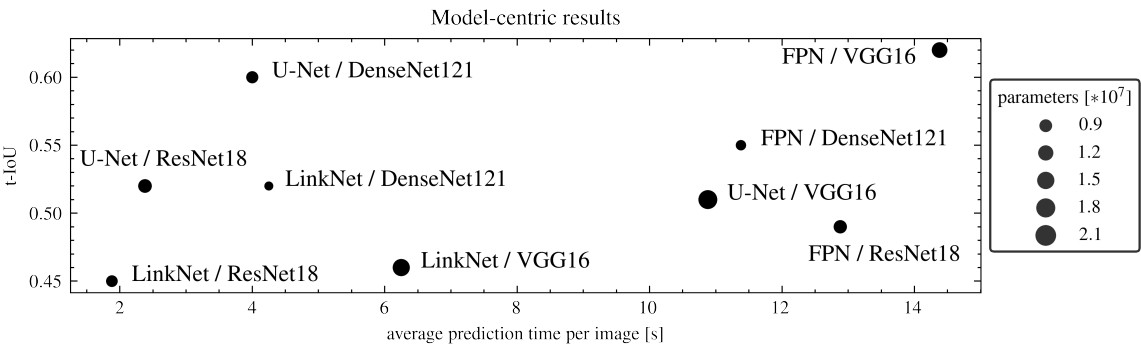

Figure 7: t-IoU scores, average prediction time and number of model parameters of each encoder-decoder model architecture

Table 1: t-IoU scores of FPN-VGG16 network architecture backbone combination for different patch sizes

| patch size | t-IoU both | t-IoU mono | t-IoU bi | training time [min] |
|---|---|---|---|---|
| 512 x 512 | **0.65** | 0.59 | **0.71** | **15.75** |
| 256 x 256 | 0.62 | **0.60** | 0.65 | 17.13 |
| 128 x 128 | 0.60 | 0.54 | 0.66 | 25.50 |
| 64 x 64 | 0.57 | 0.53 | 0.62 | 67.12 |
| 32 x 32 | 0.51 | 0.50 | 0.52 | 242.28 |

Table 2: t-IoU scores of FPN-VGG16 network architecture backbone combination with a patch size of $256 \times 256$ for different batch sizes

| batch size | t-IoU both | t-IoU mono | t-IoU bi | training time [min] |
|---|---|---|---|---|
| 32 | 0.60 | 0.55 | 0.64 | **14.64** |
| 16 | 0.58 | 0.54 | 0.63 | 14.65 |
| 8 | 0.62 | **0.60** | 0.64 | 15.93 |
| 4 | 0.61 | 0.58 | 0.64 | 17.09 |
| 2 | **0.63** | 0.59 | **0.67** | 20.10 |
| 1 | 0.62 | 0.57 | **0.67** | 25.39 |

Table 3: t-IoU scores of FPN-VGG16 network architecture backbone combination for different optimizers and losses

| loss / optimizer | t-IoU Adam | t-IoU SGD |
|---|---|---|
| categorical crossentropy (CCE) | **0.63** | 0.38 |
| categorical focal loss ($\gamma = 2$) (CF) | 0.60 | 0.42 |
| jaccard distance (JD) | **0.63** | 0.33 |

Table 4: t-IoU scores of FPN with VGG16 for different augmentations and fixed input quantity

| augmentation | sample rate | t-IoU both | t-IoU mono | t-IoU bi | training time [min] |
|---|---|---|---|---|---|
| none | 1 | 0.62 | 0.59 | 0.64 | 17.18 |
| spatial | 2 | 0.61 | 0.57 | 0.65 | 33.71 |
| pixel | 2 | 0.62 | 0.58 | 0.65 | 33.81 |
| both | 2 | 0.64 | 0.59 | **0.68** | 33.64 |
| both | 4 | **0.65** | **0.62** | **0.68** | 66.80 |
| both | 8 | 0.63 | 0.59 | 0.67 | 133.50 |
| both | 16 | 0.63 | 0.60 | 0.67 | 266.84 |

Table 5: Data-centric test **segmentation IoU** score (FPN/VGG16) for different augmentations and input quantities *(in: input, n: none, s: spatial, p: pixel, z: zoom, r: rotate)*

| in | n | s | p | s+p | z+r |
|----|------|------|------|------|------|
| 1 | 0.16 | 0.47 | 0.45 | 0.47 | 0.48 |
| 2 | 0.44 | 0.47 | 0.47 | 0.46 | 0.45 |
| 3 | 0.43 | 0.45 | 0.44 | 0.40 | 0.37 |
| 4 | 0.40 | 0.33 | 0.40 | 0.41 | 0.40 |
| 5 | 0.46 | 0.47 | 0.47 | 0.48 | 0.43 |
| 6 | 0.52 | 0.54 | 0.48 | 0.50 | 0.51 |
| 7 | 0.50 | 0.52 | 0.50 | 0.52 | 0.50 |
| 8 | 0.52 | 0.53 | 0.54 | 0.54 | 0.54 |
| 9 | 0.52 | 0.58 | 0.56 | 0.44 | 0.59 |
| 10 | 0.49 | 0.54 | 0.58 | 0.55 | 0.57 |
| 11 | 0.50 | 0.59 | 0.62 | 0.58 | 0.59 |
| 12 | 0.60 | 0.57 | 0.59 | 0.59 | 0.60 |
| 13 | 0.62 | 0.62 | 0.61 | 0.62 | 0.60 |
| 14 | 0.56 | 0.62 | 0.62 | 0.61 | 0.59 |
| 15 | 0.59 | 0.62 | 0.61 | 0.59 | 0.59 |
| 16 | 0.57 | 0.60 | 0.63 | 0.63 | **0.62** |
| 17 | 0.61 | 0.59 | 0.61 | 0.60 | 0.60 |
| 18 | 0.60 | 0.59 | 0.62 | 0.60 | 0.58 |
| 19 | 0.60 | 0.60 | 0.60 | 0.59 | 0.58 |
| 20 | 0.60 | 0.61 | 0.60 | 0.62 | 0.61 |
| 21 | 0.60 | 0.62 | 0.62 | 0.58 | 0.60 |
| 22 | 0.61 | 0.62 | 0.60 | 0.61 | 0.61 |
| 23 | 0.62 | **0.63** | **_0.64_** | 0.63 | **0.62** |
| 24 | **_0.64_** | **0.63** | 0.63 | **_0.64_** | **0.62** |

Table 6: Data-centric test **classification accuracy** (FPN/VGG16) for different augmentations and input quantities *(in: input, n: none, s: spatial, p: pixel, z: zoom, r: rotate)*

| in | n | s | p | s+p | z+r |
|----|------|------|------|------|------|
| 1 | 0.27 | 0.57 | 0.69 | 0.62 | 0.71 |
| 2 | 0.43 | 0.62 | 0.62 | 0.64 | 0.48 |
| 3 | 0.63 | 0.66 | 0.61 | 0.48 | 0.45 |
| 4 | 0.64 | 0.58 | 0.66 | 0.66 | 0.70 |
| 5 | 0.69 | 0.66 | 0.74 | 0.70 | 0.62 |
| 6 | 0.76 | 0.73 | 0.70 | 0.80 | 0.74 |
| 7 | 0.72 | 0.78 | 0.68 | 0.72 | 0.73 |
| 8 | 0.68 | 0.70 | 0.69 | 0.75 | 0.76 |
| 9 | 0.75 | 0.74 | 0.77 | 0.65 | 0.68 |
| 10 | 0.74 | 0.67 | 0.77 | 0.74 | 0.76 |
| 11 | 0.74 | 0.75 | 0.78 | 0.68 | 0.76 |
| 12 | 0.77 | 0.78 | 0.77 | 0.75 | 0.76 |
| 13 | 0.78 | **_0.82_** | 0.78 | 0.79 | 0.74 |
| 14 | 0.76 | 0.76 | 0.77 | 0.73 | 0.73 |
| 15 | 0.76 | 0.80 | 0.74 | 0.71 | 0.71 |
| 16 | 0.75 | 0.81 | 0.79 | 0.79 | 0.78 |
| 17 | 0.78 | **_0.82_** | **0.80** | 0.79 | **0.81** |
| 18 | 0.79 | 0.78 | 0.77 | 0.80 | 0.80 |
| 19 | 0.77 | 0.80 | 0.79 | **_0.82_** | 0.79 |
| 20 | 0.81 | 0.79 | 0.77 | 0.78 | 0.79 |
| 21 | 0.77 | 0.80 | 0.78 | 0.75 | 0.78 |
| 22 | **_0.82_** | 0.80 | 0.77 | 0.77 | 0.80 |
| 23 | 0.79 | 0.77 | 0.77 | 0.74 | 0.76 |
| 24 | 0.80 | 0.79 | 0.78 | **_0.82_** | 0.78 |