# OpenReview forum: "A Modular Deep Learning Pipeline for Cell Culture Analysis: Investigating the Proliferation of Cardiomyocytes"
_MIDL.io/2022/Conference — MIDL 2022_

### Official Review · Reviewer_q3Qc · 2022-01-24

**Confidence:** 4
**Preliminary Rating:** 4
**Recommendation:** Poster

**Summary:**

The authors propose a deep learning framework for the multi-cell classification of mononuclear and binuclear cardiomyocytes from confocal microscopy data. The pipeline is divided into 3 blocks: preprocessing of input data, semantic segmentation and postprocessing. Various encoder-decoder deep learning architectures are benchmarked and hyperparameter optimization is performed on the selected architecture (FPN-VGG16). The best models/parameters are identified using the IoU performance metric and segmentation results on test data are presented, both qualitatively and quantitatively. This approach could be valuable for quantitative analysis of endogeneous cardiac repair enhancement experiments.

**Strengths:**

* The authors perform extensive exploration of encoder-decoder architectures (U-Net, FPN, LinkNet) with various backbone implementations (VGG16, ResNet18, DeneNet121, InceptionV3, SEResNet18, SERwaNeXt50). The model yielding the best performance (FPN-VGG16) is then used for extensive hyperparameter optimization (notably patch size, optimizer, loss function, batch size). The authors then present their test results using the final optimized model (256 x 256 patches, Adam, CCE loss, batch size of 4). Their methodology is overall coherent and in accordance with machine learning best practices. Performance metrics chosen (IoU) are also coherent with the high class imbalance issue presented by the data.
* Although semantic segmentation of cells with deep learning is definitely not a novelty, my understanding is that the novelty here is the distinction between mononuclear and binuclear cardiomyocytes. The challenge of the problem at hand is to be able to distinguish accurately between mononuclear and binuclear (connected or separated) cells.


**Weaknesses:**

* Although the authors suggest that the multi-step pipeline is an advantage, having multiple blocks of preprocessing and postprocessing adds more complexity and can make the framework hardly adaptable and reproducible. The main advantage of end-to-end pipelines in deep learning is to avoid these complexity issues (i.e. using the ability of deep learning models to learn their own features for the task at hand instead of hand-selected pre- and postprocessing features). Making the code open access and having a clean API would strongly help mitigate this issue.
* It is unclear why the data augmentation block is located outside the semantic segmentation block in the pipeline. Typically, the data augmentation is done on the go during training (i.e. virtually). This adds to the complexity of the pipeline.


**Deanonymize Review:**

no

**Detailed Comments:**

* Recheck the spelling and grammar (e.g. in the last paragraph of page 2, is it ‘compromised’ or ‘comprised’?).
* The conclusion should mention the standout performance result (IoU of the final FPN-VGG16 model with the best hyperparameters).


**Final Rating After The Rebuttal:**

4: Weak Accept

**Justification Of The Final Rating:**

I am overall satisfied with the comments from the authors during the rebuttal. I still believe that the pipeline approach (instead of an end-to-end deep learning approach) has some flaws in this context. The best way to evaluate the approach chosen would be to make the pipeline open source and see how easily other groups can adapt it to their needs/datasets.

**Paper Type:**

methodological development

**Questions To Address In The Rebuttal:**

* It is very unconventional to consider the training-validation split ratio in the hyperparameter optimization as we already expect to have better performance and generalization with standard split ratios (i.e. 70/30, 80/20, ...). The authors themselves arrive to that conclusion on the benchmarking results of Figure 4. How do the authors justify the decision to perform hyperparameter optimization on the training-validation split ratio instead of doing, for instance, hyperparameter optimization on regularization methods such as dropout?
* Similarly, the Adam optimizer is expected to perform better for this type of application when compared to a standard SGD optimizer, but the authors still decided to include SGD on the benchmark experiments.
* The data augmentation strategies used by the authors are never explicitly defined in the paper. The ‘zoom’ and ‘rotate’ strategies are intuitive, but it is unclear how the ‘spatial’ and ‘pixel’ augmentations are performed. More clarifications are needed in the methods section.
* Also related to previous discussions, how usable would this pipeline be with different microscopy modalities or different sample stainings? How much of the authors’ preprocessing approach would need to be reimplemented to adapt to these cases?


**Special Issue:**

no

---

### Official Review · Reviewer_Z6uV · 2022-01-24

**Confidence:** 3
**Preliminary Rating:** 4
**Recommendation:** Poster

**Summary:**

This paper addresses the multi-cell segmentation and classification task in the analysis of confocal microscopy imaging data. The authors proposed a tri-part DL-based pipeline to perform such tasks and conducted extensive benchmark experiments. The authors pointed out that the proposed pipelines have the potential to benefit the automation of important medical research.

**Strengths:**

The paper addresses the need for automatic image analysis in an important area of medical research. As an application paper, the authors conducted extensive benchmarking experiments. The paper is well written and easy to follow.

**Weaknesses:**

The novelty of the paper is limited. Although this is an application paper, I feel that the overall advantage of the proposed modular framework is not adequately justified. On the dataset used by the authors, how do more basic algorithms such as the non-DL approach or end-to-end DL approach compare? The authors should at least provide more discussion on why the modular approach is preferable for practical automatic image analysis settings.

**Deanonymize Review:**

no

**Final Rating After The Rebuttal:**

4: Weak Accept

**Justification Of The Final Rating:**

The author gave some explanation to my questions. Although the novelty of the paper is limited, it may be a good reference for whose who wish to perform such tasks. Therefore I maintained my original rating.

**Paper Type:**

validation/application paper

**Questions To Address In The Rebuttal:**

I hope the authors can address my comments in the Weakness section, i.e., why is the proposed modular approach more preferable than alternative approaches. Is there a hold-out dataset to evaluate how the proposed pipeline works in practice?

**Special Issue:**

no

---

### Official Review · Reviewer_wxqf · 2022-01-25

**Confidence:** 5
**Preliminary Rating:** 2
**Recommendation:** Poster

**Summary:**

The authors propose a modular pipeline for the semantic segmentation of fixed postnatal cardiomyocytes (CM) nuclei (mononuclear and binuclear) in 2D confocal fluorescence microscopy images.
The pipeline consists of a preprocessing to obtain input patches, a deep-learning-based semantic segmentation, and a postprocessing to clean any debris from the obtained segmentations.
The authors benchmark different combinations of network architectures, hyperparameters and training data configuration.

**Strengths:**

The authors make an extensive evaluation of most of the factors affecting the performance of the deep learning based segmentation. It could be a reference for others when designing image processing experiments using deep learning.

**Weaknesses:**

The experimental design of the problem to solve is not completely clear. The authors show in Figure 1 an image with 4 channels, however, how many channels are they really using as input to the network? DAPI is labelling cells that are different from CM, which cells are those? Why do you need them to classify CM nuclei? Are you using all the channels in the semantic segmentation? In Figure 5 for example, it seems that only two channels and with different look up tables are shown.

The definition of the classes to segment is not clear to me. The authors use fixed samples to distinguish binuclear and mononuclear cells but they say that some binuclear cases could be two "separated binuclear" (Figure 6): "while others are composed of two separate nuclei, each representing one part of the same binuclear instance". When looking at the Cherry fluorescence channel, is it possible to distinguish these cases? Otherwise, would not this one be an ambiguous definition?

Although having a full benchmark of the method is strongly recommendable, some of the conclusions are too strong and not evident from the numbers shown here. Are the benchmarked architectures randomly initialised or did the authors used pretrained encoders? Specially for the case in which the networks are randomly initialised, it is highly recommendable to collect more than a single instance from each condition.

The authors claim that their method "has great potential for automating specific cell culture analyses", while the highest accuracy measure reported in the text is about 0.62 (and about 0.82 in the tables). For such a statement, I would recommend them to elaborate more on the final method's performance.

The authors conclude: "We therefore assume that an increase in training data for this specific application does not necessarily lead to improvements in performance.". A potential reason to not having seen a better performance might be the lack of independency among the samples, the ambiguity in the definition of binuclear cells or not having an adequate input size with respect to the receptive field of the network. Note that this applies both for the training and test datasets.

The description of the problem to solve and the image processing pipeline is not clear to me. The authors give low level details but the pre and post-processing are described from a high level perspective. for example the post-processing uses some morphological operations to get rid of any debris in the result but those operations are not described or stated anywhere.


**Deanonymize Review:**

no

**Detailed Comments:**


In the abstract it is said that 127 experiments are performed but then, in the main text, it is said 173. This could be a bit confusing.

Page 4: However, the to be  ... --> please rephrase it.

Figure 2: I would strongly recommend the authors to explain how are n and m being chosen. What are the z-layers?

Page 4: choose the intersection over union as a metric: does it refer to the loss function?

Page 5: "assess the quantification accuracy by detecting the contours within the respective binary layer of the semantic segmentation mask.". Does it mean that the IoU metric is only calculated along the cell border?

Page 5: "we connect binuclear instances composed of two shapes, utilizing a kd-Tree algorithm to list all pairs of contours within the range of a defined distance threshold." I would recommend citing the corresponding paper to this algorithm. Also, to which part of the workflow in Figure 2 belong this step? How would this step work if you have high density of cells that are too close to each other?

Page 5: "All models are trained with an input patch-size of 256 × 256 and 512 × 512, Adam optimizer, learning rate of 0.0001, categorical cross-entropy loss (CCE) and a batch-size of 4 for a total of 100 training epochs." This combination of parameters is less than what is shown in Figure 3.

Page 6: "the training of model configurations with larger patch sizes requires more time until convergence". In order to understand the effect of different patch sizes, it would be helpful to provide some information about the nuclei size (in pixels) and the size of the receptive field of the models. Most probably the networks are not able to properly generalise when the patch size is too small.
Likewise, the ratio of pixels belonging to the background is quite high compering to the foreground. Such imbalance may also have an effect in the performance of the model.


**Final Rating After The Rebuttal:**

3: Borderline

**Justification Of The Final Rating:**

The authors have improved the content by being more explicit and clear. Conceptually I still have the same concerns. I think the workflow is too complex and not fully reproducible for the desired task. I have updated my rating.

**Paper Type:**

methodological development

**Questions To Address In The Rebuttal:**

The pipeline described by the authors is quite complex for doing cell detection and classification. What would be the difference between the proposed pipeline and using YOLO to detect both types of cells? The signal to noise ration of the red channel looks quite high so a potential solution would be to perform cell classification with such a network and thresholding the red channel (or use stardist) to obtain the segmentation.



**Special Issue:**

no

---

### Meta-Review · Area_Chair_XT9U · 2022-02-20

**Recommendation:** Accept (Poster)
**Confidence:** 3

**Metareview:**

As pointed out by reviewers, this paper addresses the need for automatic image analysis in an important area of medical research. Although the novelty of the paper is limited, it contains extensive benchmarking experiments that could be a reference for others.

---

### Decision · Program_Chairs · 2022-02-28

Accept